# Protection of the Ovine Fetal Gut against *Ureaplasma*-Induced Chorioamnionitis: A Potential Role for Plant Sterols

**DOI:** 10.3390/nu11050968

**Published:** 2019-04-27

**Authors:** Charlotte van Gorp, Ilse H. de Lange, Owen B. Spiller, Frédéric Dewez, Berta Cillero Pastor, Ron M. A. Heeren, Lilian Kessels, Nico Kloosterboer, Wim G. van Gemert, Michael L. Beeton, Sarah J. Stock, Alan H. Jobe, Matthew S. Payne, Matthew W. Kemp, Luc J. Zimmermann, Boris W. Kramer, Jogchum Plat, Tim G. A. M. Wolfs

**Affiliations:** 1Department of Pediatrics, School of Oncology and Developmental Biology (GROW), Maastricht University, 6202 AZ Maastricht, The Netherlands; c.vangorp@maastrichtuniversity.nl (C.v.G.); i.delange@maastrichtuniversity.nl (I.H.d.L.); lilian.kessels@maastrichtuniversity.nl (L.K.); n.kloosterboer@maastrichtuniversity.nl (N.K.); luc.zimmermann@maastrichtuniversity.nl (L.J.Z.); b.kramer@maastrichtuniversity.nl (B.W.K.); 2Department of Surgery, School for Nutrition, Toxicology and Metabolism (NUTRIM), Maastricht University, 6202 AZ Maastricht, The Netherlands; w.vangemert@maastrichtuniversity.nl; 3Cardiff University School of Medicine, Cardiff CF10 3AT, Wales, UK; spillerb@cardiff.ac.uk; 4Maastricht Multimodal Molecular Imaging Institute (M4I), Maastricht University, 6202 AZ Maastricht, The Netherlands; f.dewez@maastrichtuniversity.nl (F.D.); b.cilleropastor@maastrichtuniversity.nl (B.C.P.); r.heeren@maastrichtuniversity.nl (R.M.A.H.); 5Cardiff School of Health Sciences, Cardiff Metropolitan University, Cardiff CF14 4XN, UK; mbeeton@cardiffmet.ac.uk; 6MRC Centre for Reproductive Health, Queen’s Medical Research Institute, University of Edinburgh, Edinburgh EH16 4TJ, UK; sarah.stock@ed.ac.uk; 7Division of Neonatology/Pulmonary Biology, The Perinatal Institute, Cincinnati Children’s Hospital Medical Center, University of Cincinnati, Cincinnati, OH 45229, USA; alan.jobe@cchmc.org; 8Division of Obstetrics and Gynecology, School of Medicine, The University of Western Australia, Crawley WA 6009, Australia; matthew.payne@uwa.edu.au; 9School of Women’s and Infant’s Health, The University of Western Australia, Crawley WA 6009, Australia; matthew.kemp@uwa.edu.au; 10Department of Nutrition and Movement Sciences, School for Nutrition and Translational Research in Metabolism (NUTRIM), Maastricht University, 6202 AZ Maastricht, The Netherlands; j.plat@maastrichtuniversity.nl; 11Department of Biomedical Engineering (BMT), School for Cardiovascular Diseases (CARIM), Maastricht University, 6202 AZ Maastricht, The Netherlands

**Keywords:** chorioamnionitis, *Ureaplasma parvum*, ovine, plant sterols, β-sitosterol, campesterol, fetal inflammatory response syndrome, intestinal inflammation, intestinal lipidome

## Abstract

Chorioamnionitis, clinically most frequently associated with *Ureaplasma*, is linked to intestinal inflammation and subsequent gut injury. No treatment is available to prevent chorioamnionitis-driven adverse intestinal outcomes. Evidence is increasing that plant sterols possess immune-modulatory properties. Therefore, we investigated the potential therapeutic effects of plant sterols in lambs intra-amniotically (IA) exposed to *Ureaplasma*. Fetal lambs were IA exposed to *Ureaplasma parvum* (*U. parvum*, UP) for six days from 127 d–133 d of gestational age (GA). The plant sterols β-sitosterol and campesterol, dissolved with β-cyclodextrin (carrier), were given IA every two days from 122 d–131 d GA. Fetal circulatory cytokine levels, gut inflammation, intestinal injury, enterocyte maturation, and mucosal phospholipid and bile acid profiles were measured at 133 d GA (term 150 d). IA plant sterol administration blocked a fetal inflammatory response syndrome. Plant sterols reduced intestinal accumulation of proinflammatory phospholipids and tended to prevent mucosal myeloperoxidase-positive (MPO) cell influx, indicating an inhibition of gut inflammation. IA administration of plant sterols and carrier diminished intestinal mucosal damage, stimulated maturation of the immature epithelium, and partially prevented *U. parvum*-driven reduction of mucosal bile acids. In conclusion, we show that β-sitosterol and campesterol administration protected the fetus against adverse gut outcomes following UP-driven chorioamnionitis by preventing intestinal and systemic inflammation.

## 1. Introduction

Preterm birth is the leading cause of neonatal mortality and morbidity, and it accounts for 35% of neonatal deaths worldwide [1]. Annually, 15 million children are born prematurely, an incidence that is still increasing [2]. Chorioamnionitis, defined as inflammatory cell infiltration of fetal membranes, is an important cause of preterm birth [3]. During chorioamnionitis, contaminated amniotic fluid (AF) is swallowed by the fetus, consequently infecting the gastrointestinal system. We have previously studied the effects of chorioamnionitis on the fetal gut in a translational ovine chorioamnionitis model that is of relevance to human pathology because of the close resemblance between the developmental biology and physiology of human and ovine fetuses [4]. In this model, intra-amniotic (IA) infections with micro-organisms and inflammatory mediators (lipopolysaccharide (LPS)) can induce gut inflammation and subsequent gut injury and developmental alterations [5,6]. Accordingly, chorioamnionitis is associated with an increased risk of postnatal intestinal pathologies, such as necrotizing enterocolitis (NEC), which is known for its high mortality and morbidity rates [7,8,9]. In addition, chorioamnionitis is associated with the fetal inflammatory response syndrome (FIRS), which is characterized by increased IL-6 concentrations in fetal blood and is an independent risk factor for severe neonatal morbidity [10]. 

Currently, no treatments are able to prevent intestinal inflammation and its sequelae following chorioamnionitis or to consequently prevent postnatal intestinal disorders [9]. Plant sterols, also known as phytosterols, are a dietary component derived from vegetable oils, fruits, nuts, and grains [11]. They were first evaluated in the atherosclerosis field for their cholesterol-lowering effects [12]. In addition, their immune-regulatory potential has been increasingly recognized [13]. Recently, these plant sterols were shown to possess anti-inflammatory properties in the context of intestinal inflammatory diseases such as inflammatory bowel disease [14,15,16]. Interestingly, this was recently confirmed in a small pilot study in fetal lambs, in which intestinal inflammation and mucosal injury following IA LPS exposure were prevented by plant sterol treatment [17]. 

This pilot study prompted us to conduct the current in-depth study, in which chorioamnionitis was induced by viable *U. parvum* serovar 3, the micro-organism most frequently associated with chorioamnionitis. UP, a mycoplasma present in the female urogenital tract [18], inflicts a milder hit than LPS does. Nevertheless, UP colonization in preterm infants has been found to increase NEC incidence twofold, with a gestational age-adjusted odds ratio (OR) of 2.47 (95% CI, 1.13–5.43) [19,20], making UP a clinically relevant stressor.

We IA-administered a mixture of β-sitosterol and campesterol, the two most common plant sterols in nature [11], as a treatment prior to IA injection of UP serovar 3. The effects of plant sterol treatment in the context of UP-induced chorioamnionitis were investigated by studying circulatory cytokine levels, fetal gut inflammation, intestinal delivery of the plant sterols, and intestinal injury and maturation.

## 2. Materials and Methods

### 2.1. Experimental Design

The animal studies were approved by the Animal Ethics Committee of the University of Western Australia (Perth, Australia), and the National Research Council’s guide for the care and use of laboratory animals was followed. Time-mated Merino cross-breed ewes with singleton fetuses were randomly assigned to the 6 study groups. A total number of 50 animals were used, based on a power analysis with intestinal inflammation as the primary outcome. 

Following drop-outs, the group size for data analyses was six to seven animals per group (Figure 1). Animals were group-housed with a 12-h dark/light cycle and had ad libitum access to food and water. Animal welfare was assessed daily by qualified personnel. The experiment was reported to be in compliance with the ARRIVE (Animal Research: Reporting of In Vivo Experiments) guidelines [21].

To investigate whether plant sterols can be used as a nutritional intervention in utero to improve fetal outcomes in the context of chorioamnionitis, we deliberately chose IA administration, since the pharmacokinetics of plant sterols in sheep are to date unclear. In addition, several findings have supported the concept that plant sterol availability to the fetus is determined by the maternal diet. First, plant sterols are not synthesized by animals and humans and are exclusively gained through the diet [22]: The natural presence of plant sterols in AF thus implies that maternal diet-derived plant sterols reach the AF [23,24]. Second, the plant sterol transporter NPC1L1 is present in the human placenta [22,25]. Finally, a meta-analysis from Ras et al. reported that oral plant sterol administration increases plasma plant sterol concentrations [26]. Therefore, increasing plant sterol levels in the maternal diet is clinically the most relevant route of administration.

Therefore, we enriched the AF of pregnant ewes as a model for AF enrichment following maternal oral plant sterol intake. A mixture of β-sitosterol (70%) and campesterol (30%) (total of 0.6 mg/mL) dissolved in a carrier, 18% 2-hydroxypropyl-β-cyclodextrin (H107, Sigma Aldrich, St. Louis, MO, USA) in saline, carrier in saline, or saline alone was injected IA at 122 d of gestational age (GA), five days before IA UP injection. This dosage was chosen to supplement endogenous plant sterol concentration, starting with a twofold increase. Since the plant sterol mixture was hydrophobic, the carrier β-cyclodextrin was essential in dissolving the plant sterols (transport). Animals were injected every two days with additional IA injections containing saline or a plant sterol mixture until 131 d GA. AF samples were collected every 2 days from 122 d GA until preterm delivery at 133 d GA. Viable *U. parvum* serovar 3 (107 color changing units (CCUs)) or saline was given IA at 127 d of gestational age (GA). UP was grown in vitro and injected into the AF under ultrasound guidance as reported previously [27]. AF sampled at 133 d GA was cultured for UP enumeration [28]. Colonization of UP was not detected in the AF of controls. Fetuses were delivered preterm by caesarean section at 133 d GA (150 d ~ term), which is (with regard to the gut) comparable to 33–34 weeks of human gestation. Lambs were euthanized directly after delivery by an intravenous injection of pentobarbitone (100 mg/kg, Valaberb, Pitman-Moore, Australia). Distal ileum and blood samples for obtaining plasma were collected postmortem. The investigators involved in the data analyses were blinded to treatment allocation.

### 2.2. Antibodies

The following antibodies were used: polyclonal rabbit antibody against human myeloperoxidase (MPO) (A0398, Dakocytomation, Glostrup, Denmark) and cluster of differentiation 3 (CD3 (A0452, Dakocytomation, Glostrup, Denmark), monoclonal rabbit antibody against human fork head box P3 (FoxP3) (clone eBio7979, 14-7979-82, eBioscience, San Diego, CA, USA), ovine interleukin-6 (IL-6) (MAB1004, Millipore, Darmstadt, Germany), and ovine IL-8 (MAB1044 Millipore, Darmstadt, Germany). Antibodies against intestinal fatty acid binding protein (I-FABP) were kindly provided by the Department of Surgery, Maastricht University Medical Centre, the Netherlands. Secondary antibodies were the following: biotin-conjugated rabbit antimouse (E0413, DakoCytomation, Glostrup, Denmark), swine antirabbit (E0353, DakoCytomation, Glostrup, Denmark), and peroxidase-conjugated goat antirabbit (111-035-045, Jackson, West Grove, PA, USA). Detection antibodies against IL-6 (AB1839, Millipore, Darmstadt, Germany) and IL-8 (AB1840, Millipore, Darmstadt, Germany) were used.

### 2.3. ELISA

Plasma I-FABP concentrations were assessed by ELISA. A high-sensitivity ELISA kit was kindly provided by the Department of Surgery, Maastricht University Medical Centre, the Netherlands. The protocol used was as described previously [29]. Circulatory IL-6 and IL-8 concentrations were measured to assess whether systemic inflammation was present as previously described [30]. Briefly, a 96-well plate (ELISA 96-well, Greiner Bio One) was precoated with 100 μl of IL-6 (5 µg/mL) or IL-8 (5 µg/mL) antibodies overnight at 4 °C. After incubation and washing, nonspecific binding sites were blocked. For the standard curve, protein standards were prepared by two-step serial dilutions of recombinant IL-6 or IL-8 (ImmunoChemistry Technologies, Bloomington, MN, USA). Diluted plasma samples and protein standards were added in duplicate and incubated for 2 h at 37 °C. After washing, detection antibodies were added for one hour at room temperature (RT). Detection antibodies against IL-6 or IL-8 were detected using a peroxidase-conjugated antibody and 3,3′,5,5′-tetramethylbenzidine (TMB) substrate solution (Sigma Aldrich, St. Louis, MO, USA).

### 2.4. Immunohistochemistry

We fixed distal ileum in 4% paraformaldehyde, embedded it in paraffin, and cut 4-um sections with a Leica RM2235 microtome to perform immunohistochemistry studies. Intestinal mucosal damage and morphological changes were assessed using hematoxylin and eosin staining (H&E). Per animal, a histological score was assigned to the intestinal section by 2 blinded investigators. The following scoring system (from no injury to severe injury) was developed to describe the severity of histological injury: no damage, mild damage (disrupted epithelial lining, but no loss of enterocytes), moderate damage (disrupted epithelial lining, moderate enterocyte loss from the villus tips), or severe damage (a disrupted epithelial lining, abundant enterocyte loss from villus tips). 

Sections were stained for CD3, FoxP3, MPO, and I-FABP as described previously [6]. Briefly, endogenous peroxidase activity was blocked with 0.3% H_2_O_2_ in phosphate-buffered saline. For CD3, antigen retrieval was performed by boiling in 10 mM of sodium-citrate buffer (pH 6.0) for 10 min. Blocking of nonspecific binding sites was performed with normal goat serum (MPO) or bovine serum albumin (CD3, FoxP3, I-FABP) for 30 min at room temperature. Subsequently, slides were incubated with the primary antibody of interest for one hour at room temperature (MPO) or overnight at 4 °C (CD3, FoxP3, I-FABP). After washing, slides were incubated with biotin-conjugated (CD3, FoxP3) or peroxidase-conjugated secondary antibodies (MPO, I-FABP). MPO and I-FABP positivity was visualized with 3-amino-9-ethylcarbazole (AEC, Sigma Aldrich, St. Louis, MO, USA). CD3- and FoxP3-positive cells were detected using nickel-3,3′-diaminobenzidine (Sigma Aldrich, St. Louis, MO, USA). Nuclei were counterstained with hematoxylin (MPO and I-FABP) or with nuclear fast red (CD3 and FoxP3). The number of MPO-, CD3-, and Foxp3-positive cells were counted per high power fields (100x) using a light microscope (Leica Microsystem CTG, type DFC295) and ImageJ (1.52b software, National Institutes of Health, Bethesda, MD, USA). For MPO, the number of MPO+ cells present in the villi was determined separately, and the percentage of MPO+ cells in the villi compared to the total MPO+ cell count was determined. The average value of five representative high power fields was used for the analyses. 

### 2.5. Gas–Liquid Chromatography–Mass Spectrometry (GC-MS)

Plant sterols (β-sitosterol and campesterol), cholesterol and cholesterol precursor (desmosterol and lathosterol) concentrations in plasma (lathosterol), and AF samples (demosterol and lathosterol) were measured using gas chromatography–mass spectrometry as described previously [12]. 

### 2.6. Matrix-Assisted Laser Desorption Ionization Mass Spectrometry Imaging

To study the distribution and changes in intestinal lipid profiles in more detail, molecular analysis of the distal ileum (*n* = 3 per group) was performed by matrix-assisted laser desorption ionization mass spectrometry imaging (MALDI-MSI). After sampling, the intestinal tissues of all treatment groups (control, sterol + carrier, carrier, 6-d UP, 6-d UP + sterol + carrier, and 6-d UP + carrier) were frozen in liquid nitrogen, and cryo-sections of 10 µm were cut in a cryostat (Leica CM3050S, Amsterdam, The Netherlands). The tissue sections were deposited on indium tin oxide conductive slides (Delta Technologies, Loveland, CO, USA) and stored at −20 °C. A matrix solution consisting of norharman (7 mg/mL) in 2:1 chloroform/methanol was sprayed on top of the tissue sections by using a TM-Sprayer M3 (HTX Technologies, Carrboro, NC, USA). The imaging experiments were performed with a Bruker RapifleX MALDI Tissuetyper in reflectron mode (Bruker Daltonik GmbH, Bremen, Germany) at a raster size of 80 µm. Data were acquired in positive ion mode in the mass range *m*/*z* 300–1600 and in negative ion mode in the mass range *m*/*z* 350–1600. Each tissue section was used for both polarities, with an offset of 40 µm in the *X*/*Y* directions. High spatial resolution experiments were performed using 5 mg/mL of α-cyano-4-hydroxycinnamic acid (Sigma Aldrich, St. Louis, MO, USA) in 70% acetonitrile and 0.2% trifluoroacetic acid and were sprayed with a SunCollect sprayer (SunChrom). Data were acquired in positive mode with a MALDI HDMS SYNAPT G2-Si mass spectrometer (Waters, Manchester, UK) with a modified MALDI source unit, which achieves a laser spot 15 µm in diameter at a spatial resolution of 20 µm [31]. Lipid identifications were obtained using a high-mass resolution MALDI-MSI Orbitrap Elite Hybrid Ion trap mass spectrometer (Thermo Fisher Scientific, Bremen, Germany). Tandem Mass Spectrometry (MS/MS) spectra (Appendix A) were submitted to the ALEX123 lipid database (http://alex123.info/ALEX123/MS.php). To study differences between all conditions, principal component analyses (PCAs) and discriminant analysis (DA) were performed. Both the PCAs and DA were performed by using an in-house built ChemomeTricks toolbox for MATLAB version 2014a (MathWorks, Natick, MA, USA). For analyzing lipid patterns in the intestinal epithelium, luminal lipid signals were excluded from the data analyses. Only for an analysis of β-cyclodextrin presence were luminal signals included.

### 2.7. Statistics

Statistical analyses were performed using GraphPad Prism software (version v6.0, GraphPad Software Inc., La Jolla, CA, USA). Data are presented as the median with an interquartile range (IQR) for all read-outs, except for the damage score, where only the median is presented. A nonparametric Kruskal–Wallis test followed by Dunn’s post hoc test was used to analyze significant differences between the groups. Differences were regarded as statistically significant at *p* < 0.05: *p*-values ≤ 0.1 were interpreted as biologically relevant, as previously described [32].

## 3. Results

### 3.1. Systemic and Intestinal Inflammation

#### 3.1.1. Systemic Inflammation

Plant sterol treatment before and during UP exposure significantly lowered systemic IL-6 concentrations compared to UP-exposed animals (*p* < 0.05) (Figure 2). No significant changes were observed in IL-8 plasma concentrations among the several treatment groups (Appendix A).

#### 3.1.2. Intestinal inflammation 

The percentage of MPO-positive cells that migrated into the villi in the distal ileum increased significantly after 6 d of UP exposure (*p* < 0.05) compared to control animals (Figure 3A,B,D). This increase tended to be reduced (*p* = 0.10) by plant sterols (Figure 3C,D). The number of FoxP3+ and CD3+ cells did not differ with statistical significance between the experimental groups (Figure 4A,B). The FoxP3+/CD3+ ratio tended to be reduced in fetuses exposed to UP (*p* = 0.06), which was not restored by plant sterol treatment. Remarkably, plant sterol treatment alone also tended to reduce the FoxP3+/CD3+ ratio (*p* = 0.09) (Figure 4C).

### 3.2. UP and Sterol Concentrations in Amniotic Fluid

UP infection was confirmed by the presence of UP in the AF of ewes injected with UP (Appendix A). No statistically significant differences were observed in UP CCUs between the three treatment groups. No UP CCUs were present in the AF of animals that were not injected with UP. 

IA concentrations of β-sitosterol in the sterol group increased with repeated doses and were statistically significantly (*p* < 0.05) elevated compared to controls at 133 d GA, indicating the accumulation of β-sitosterol over time with IA delivery (Figure 5A). Here, β-sitosterol concentrations were not statistically significantly increased in the 6-d UP group treated with plant sterols at 133 d GA. The AF campesterol concentration tended to be increased (*p* = 0.09) at 133 d GA in ewes that were treated with IA plant sterols compared to control ewes (Figure 5B). Campesterol was not statistically significantly elevated in the AF of ewes that were exposed to UP and plant sterols at 133 d GA. No statistically significant differences were found in lathosterol, desmosterol, and cholesterol concentrations in the AF between treatment groups (Appendix A).

### 3.3. Sterol Concentrations in Fetal Plasma

Fetal plasma concentrations of β-sitosterol and campesterol in the sterol-supplemented groups were not significantly elevated at 133 d GA compared to the other groups (Figure 6A,B). Lathosterol concentrations were statistically significantly decreased after 6 d of UP exposure with plant sterol treatment (*p* < 0.05) and tended to be decreased after 6 d of UP exposure alone (*p* = 0.10) compared to the control animals (Figure 6C). The changes in plasma lathosterol concentrations did not result in altered plasma cholesterol concentrations in the different experimental groups (Figure 6D).

### 3.4. Intestinal Damage and Gut Maturation

Enterocyte loss at the villi tips in H&E stained ileal tissue, indicating moderate to severe intestinal damage, was found in five out of seven of the 6-d UP-exposed lambs, compared to one out of six controls (Figure 7A,B; Appendix A). Intestinal damage was partly blocked by the administration of plant sterols (only two out of six had moderate to severe damage) and the carrier (three out of seven had moderate to severe damage) (Figure 7C,D; Appendix A). No changes were observed in fetal plasma concentrations of I-FABP (Appendix A), a small protein present in the cytoplasm of mature enterocytes. Furthermore, the sizes of enterocyte vacuoles were increased in all experimental groups when compared to control animals (Figure 7A–D). 

To assess enterocyte maturation, the distribution of I-FABP was analyzed in the distal ileum. I-FABP expression was altered in the 6-d UP-exposed fetuses compared to controls, as expression was more localized in the intestinal crypts (Figure 8A,B). This disturbed I-FABP distribution was partially restored by carrier administration (Figure 8C) as well as by plant sterol treatment (Figure 8D).

### 3.5. Changes in Lipid Profiles in the Fetal Gut

MALDI-MSI was used for the following reasons: (1) to study changes in the morphological appearance of the vacuolated enterocytes after plant sterol treatment, (2) as an additional read-out for inflammation, and (3) to confirm that plant sterols were delivered in the fetal intestinal lumen following IA plant sterol administration. Interestingly, lipid composition was changed after 6 d UP exposure compared to the other treatment groups and was characterized by an increased presence of different phosphatidylcholines (PCs) and sphingomyelins (SMs), as the first discriminant function (DF1) shows (Figure 9A–C). A clear accumulation of *m*/*z* 782.58 PC 34:1 [M + Na]^+^ was detected in the 6-d UP group compared to the control situation (Figure 9C). Furthermore, *m*/*z* 810.60 PC 36:1 [M + Na]^+^ and *m*/*z* 725.57 SM 34:1 [M + Na]^+^ were also enriched in the 6-d UP group lipid profile (Figure 9B). These UP-induced changes of the intestinal lipidome were partially reduced by plant sterols and/or the carrier (Figure 9D–G). The carrier was characterized by *m*/*z* 1331.52 [M + Na]^+^ and was confirmed by tandem MS [33]. Other peaks related to the carrier, such as *m*/*z* 1273.48, *m*/*z* 1389.55, and *m*/*z* 1447.59, were also detected in the intestinal epithelium of the sterol + carrier, carrier, UP + sterol + carrier, and UP + carrier groups (Figure 9F,G). The presence of the carrier was mainly detected in the intestinal lumen in all carrier and plant sterol groups (sterol + carrier, carrier, 6-d UP + sterol + carrier, and 6-d UP + carrier), but not in the control and 6-d UP groups (Figure 10), showing intraluminal delivery and subsequent uptake in the distal ileum. 

Lipid species, including phosphatidylinositols (PIs) and bile acids (BAs), were identified in the negative ion mode by using MALDI-MSI (Figure 11). In line with the results in the positive ion mode, the most profound lipidome changes were detected in the UP-exposed animals (Figure 11A). More precisely, a higher abundance of *m*/*z* 885.58 PI 38:4 [M−H]^−^ and a strong reduction of dihydroxy BA *m*/*z* 498.29 taurodeoxycholic acid/taurochenodeoxycholic acid [M−H]^−^ (TCDA/TCDCA) and trihydroxy BA *m*/*z* 514.29 [M−H]^−^ taurocholic acid (TCA) following UP infection were found, which were indicative of a proinflammatory environment and a reduced BA status (Figure 11A,B). These UP-induced lipidome and BA changes detected in the negative ion mode were partially prevented by plant sterol and/or carrier administration (Figure 11C–F).

## 4. Discussion

An important finding was that IA plant sterol administration decreased the fetal systemic inflammatory response (circulatory IL-6 levels) during UP-driven chorioamnionitis. This corresponds with a study from Bouic et al. showing that a systemic inflammatory response following excessive exercise (increased serum IL-6 concentrations) was prevented by supplementing ultramarathon athletes with capsules containing a mixture of plant sterols [34,35]. This decrease in systemic inflammation was supported by work from Nashed et al., who showed that splenocytes cultured from plant sterol-treated apo E-knockout mice reduced the production of IL-6 upon stimulation with LPS [36].

In addition, our data suggest that IA plant sterol treatment partially modulates fetal intestinal inflammation in response to UP infection. This result is supported by a recent pilot study in which we showed that IA plant sterol administration completely prevented mucosal neutrophil infiltration in ovine fetuses IA exposed to LPS instead of UP [17]. In this study, IA LPS exposure provoked excessive inflammation and severe epithelial injury. Such severe injury might be a prerequisite to better detect the protective effects of plant sterols. In an earlier study of UP-induced chorioamnionitis, we reported that intestinal inflammation after IA UP exposure was characterized by an imbalance between regulatory T-cells (Treg) and effector T-cells [37]. This was confirmed in the current study, where we observed a trend toward a reduced FoxP3+/CD3+ ratio with UP exposure. In our experiments, IA plant sterol administration tended to decrease the FoxP3+/CD3+ ratio, indicating that IA plant sterol administration may suppress the number of intestinal Tregs. Other studies have also reported the effects of plant sterols on Treg cell numbers, although both increased and decreased Treg numbers have been observed. More precisely, Te Velde et al. found an increase in Treg numbers and a decrease in total T-cell numbers in the colon in a murine T-cell transfer colitis model after dietary plant sterol enrichment [14]. De Smet et al. showed that oral administration of plant sterols reduced both Treg numbers and T-cell numbers in the jejunum of healthy volunteers with a concomitant reduction in T-cell-associated gene expression [38]. This variation illustrates that the effect of plant sterols on T-cells is determined by the underlying pathophysiology, intestinal region, timing of the intervention, duration, and nature of the inflammatory stimulus and developmental stage.

Since plant sterols are known to influence cholesterol metabolism, which may be relevant to fetal/neonatal growth and development [39], cholesterol and cholesterol precursor concentrations in AF and plasma were studied. No changes were seen in AF and fetal plasma in the plant sterol group compared to the control group, indicating that plant sterol administration did not disturb systemic cholesterol concentrations or cholesterol synthesis in the fetus. The reason for the increased size of enterocyte vacuoles following plant sterol and carrier supplementation remains unknown. Interestingly, UP administration did reduce endogenous cholesterol synthesis, as reflected by reduced fetal plasma lathosterol concentrations. The mechanisms by which IA UP exposure can influence endogenous cholesterol synthesis are currently unclear. The disturbance of endogenous cholesterol synthesis was not prevented by plant sterol administration.

In this study, we found an enrichment of phospholipids (i.e., phosphatidylcholines, sphingomyelins, and phosphatidylinositol) [40,41,42,43] present in the cell membrane and involved in diverse cell signaling processes following IA UP exposure. Interestingly, similar changes in the cerebral lipodome were found by Gussenhoven et al. in ovine fetuses after two days of IA LPS exposure, and these changes were indicative of an inflammation-induced “diseased” lipid profile [44]. Although such lipid disturbances have been associated with detrimental effects on the central nervous system, immune system, and skeletal muscles [45], the significance for the intestine is still unknown [46]. 

Besides changes in phospholipids, BAs were reduced in UP-exposed animals. BAs are critical to the facilitation of dietary lipid absorption, intestinal antimicrobial defense [47,48], and regulation of gastrointestinal motility [49,50]. A reduction of intestinal BA following antenatal UP exposure might therefore subvert lipid malabsorption, microbial defense, and intestinal motility, which are known risk factors for NEC development [51,52,53]. In line with this theory, an increased incidence of NEC has been associated with UP infections [19,20].

Interestingly, in this study, plant sterols and/or the carrier reduced mucosal damage and stimulated the maturation of enterocytes. Based on these data, we cannot define to what extent plant sterols and/or the carrier are responsible for protective gut barrier/maturational effects, but several findings have indicated that the effects are plant sterol-specific. First, as we recently showed, plant sterol administration, but not the carrier, prevented severe damage to the fetal gut following LPS-induced chorioamnionitis [17]. Second, plant sterols prevent systemic inflammation, which is a known inducer of gut integrity loss.

In this short-term proof of concept study, repetitive IA plant sterol injections induced a gradual increase in AF plant sterol concentrations. This ultimately led to a 10-fold increase, which is likely higher than can be expected following maternal oral administration of plant sterols. Further studies are thus needed to explore the potential beneficial effects of a maternal oral nutritional intervention with β-sitosterol and campesterol on the fetus over time. 

In our study, we found therapeutic effects of the carrier β-cyclodextrin that were milder, but which recapitulated the effects of β-cyclodextrin + plant sterols. Overlapping working mechanisms of both plant sterols and β-cyclodextrin may explain these findings: β-cyclodextrin has previously been shown to exert anti-inflammatory effects in the field of atherosclerosis [54,55,56], mediated (among other things) by an increase in liver-X-receptor (LXR) target gene expression [56]. Interestingly, plant sterols are also known activators of LXRs [57], and LXR signaling has been implicated in playing a role in experimental colitis and inflammatory bowel disease [58,59]. As more prominent and exclusive effects of plant sterols are seen when compared to carrier-only-treated animals, it is tempting to speculate that plant sterols target LXR-independent pathways, which remains to be further elucidated. 

## 5. Conclusions

In conclusion, we showed in this proof of concept study that IA β-sitosterol and campesterol administration prevented fetal systemic and intestinal inflammation. Moreover, the combined results from the LPS- and UP-induced chorioamnionitis models showed that plant sterols have the potential to prevent intestinal mucosal injury. Future studies are required to investigate the pharmacokinetics of plant sterols in the fetus following maternal intake during human pregnancy to further dissect the role of plant sterols and the carrier in the protection of the fetal gut and ultimately unravel the clinical potential of plant sterols in the perinatal context.

## Figures and Tables

**Figure 1 nutrients-11-00968-f001:**
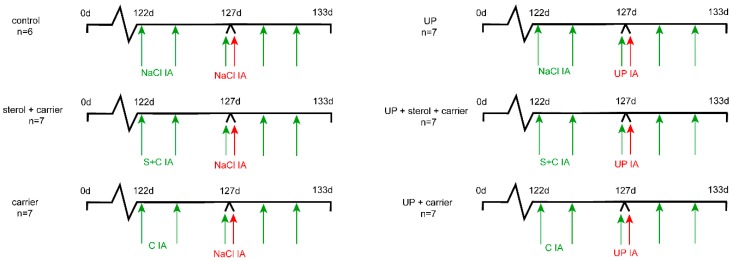
Experimental design. Animals were randomly assigned to six study groups of six to seven animals. Plant sterols dissolved with a carrier (C) (β-cyclodextrin) were given by intra-amniotic (IA) injection at 122 d of gestational age (GA), before onset of chorioamnionitis. Plant sterol injections were repeated every 2 days until 131 d GA, followed by premature delivery at 133 d GA. *U. parvum* serovar 3 (10^7^ color changing units (CCUs)) was given by IA injection at 127 d GA to induce chorioamnionitis. Control groups had saline injections. Two groups received the carrier (β-cyclodextrin) IA without plant sterols to assess the carrier separately from the plant sterols. Abbreviations: C, carrier; IA, intra-amniotic; S, plant sterols; UP, *U. parvum*.

**Figure 2 nutrients-11-00968-f002:**
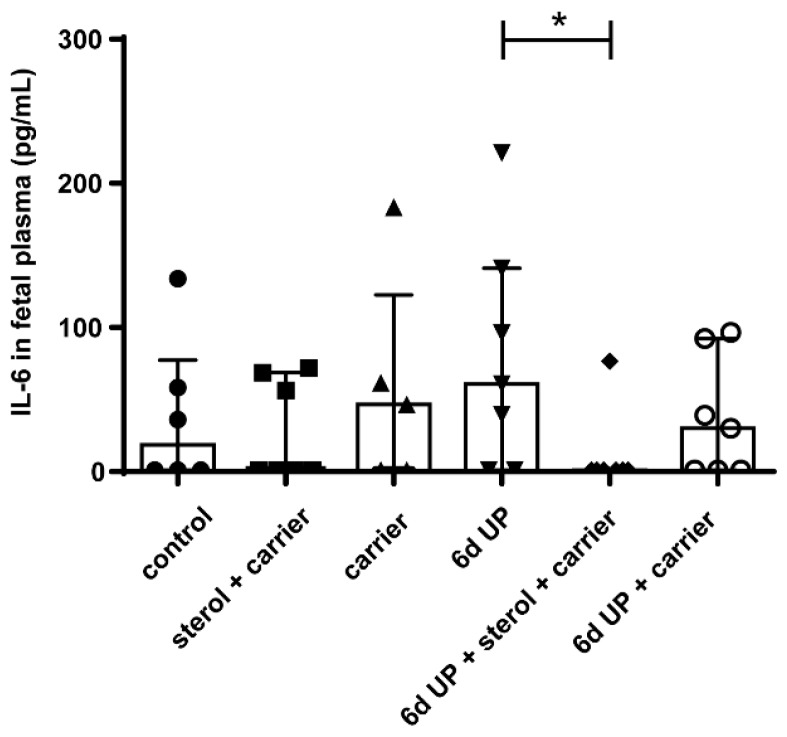
Circulatory IL-6 levels in fetuses of 133 d GA. Plant sterols significantly inhibited circulatory IL-6 levels. * *p* < 0.05. Abbreviations: GA, gestational age; UP, *U. parvum*.

**Figure 3 nutrients-11-00968-f003:**
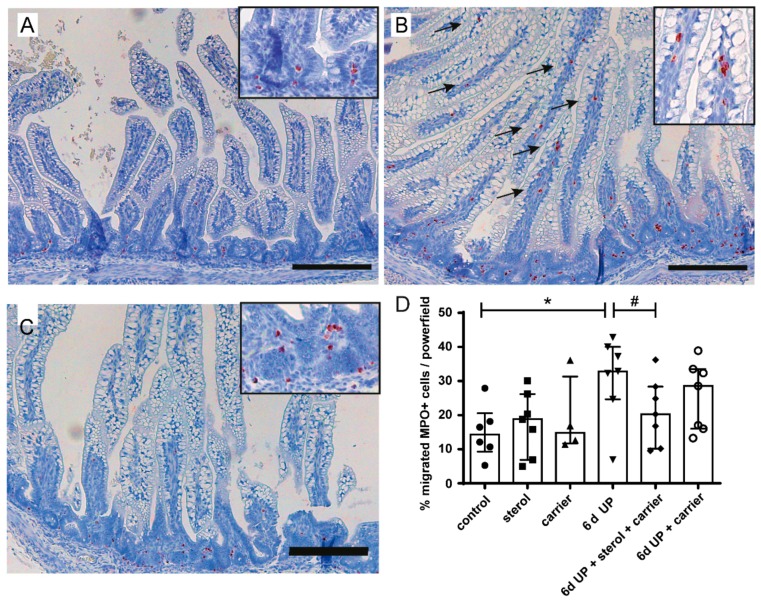
Immunohistochemical distribution of MPO-positive cells. (**A**) In control fetuses, MPO-positive cells were predominantly located in the lower crypt region. (**B**) After 6 d of UP exposure, the percentage of migrated MPO-positive cells (indicated by black arrows) increased compared to control fetuses. (**C**) An influx of MPO-positive cells in the villi tended to be prevented after treatment with plant sterols. (**D**) The total number of MPO-positive cells and the number of migrated MPO-positive cells were counted per high-power field, and the mean value of the count in five representative high-power fields is given. Scale bar indicates 200 µm. * *p* < 0.05; # 0.05 < *p* < 0.10. Abbreviations: MPO, myeloperoxidase; UP, *U. parvum*.

**Figure 4 nutrients-11-00968-f004:**
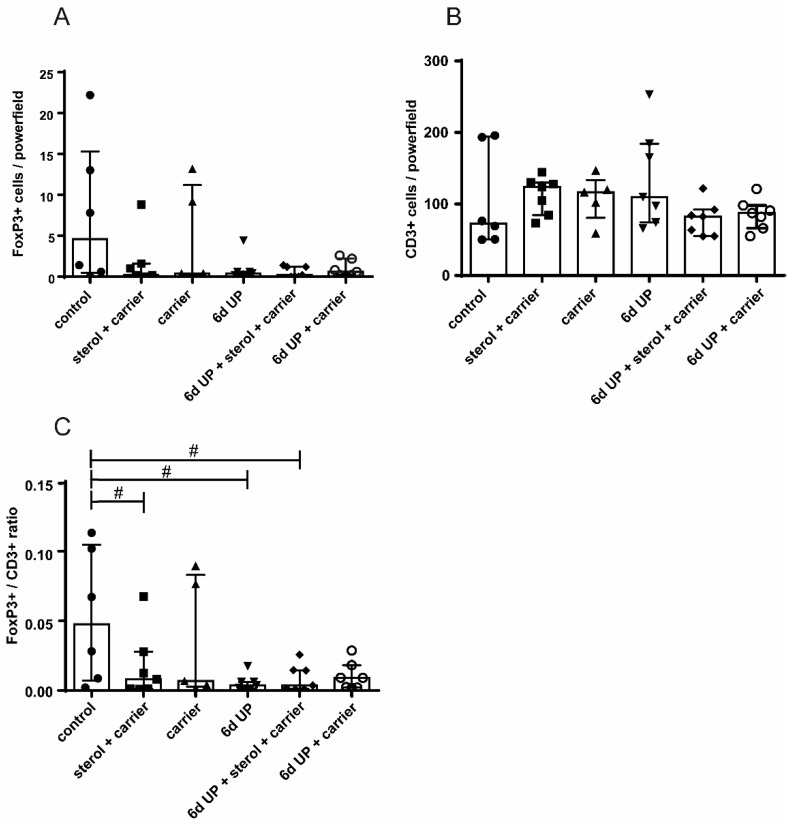
The number of CD3-positive cells, FoxP3-positive cells, and the FoxP3+/CD3+ ratio in the distal ileum of preterm lambs. (**A**) FoxP3-positive cells and (**B**) CD3-positive cells are displayed for all treatment groups. No significant differences were observed. (**C**) The FoxP3+/CD3+ ratio tended to be reduced in fetuses of the sterol + carrier, 6-d UP, and 6-d UP + sterol + carrier groups. The number of CD3+ and FoxP3+ cells were counted per high-power field, and the mean value of the count in five representative high-power fields is given. # 0.05 < *p* < 0.10. Abbreviations: CD3, cluster of differentiation 3; FoxP3, fork head box P3; UP, *U. parvum*.

**Figure 5 nutrients-11-00968-f005:**
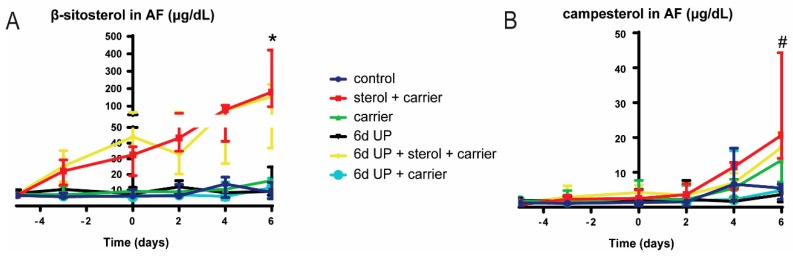
AF concentrations of β-sitosterol and campesterol. AF samples were taken every two days from 122 d GA until preterm delivery at 133 d GA. (**A**) AF concentrations of β-sitosterol were significantly increased at day 11 in the sterol + carrier group. (**B**) AF concentrations of campesterol tended to be increased at day 11 in the sterol + carrier group. Day 0 (122 d GA) is the start of plant sterol treatment, day 5 (127 d GA) is the day of intra-amniotic UP injection, and day 11 (133 d GA) is the moment of preterm delivery. * *p* < 0.05; # 0.05 < *p* < 0.10. Abbreviations: AF, amniotic fluid; GA, gestational age; UP, *U. parvum*.

**Figure 6 nutrients-11-00968-f006:**
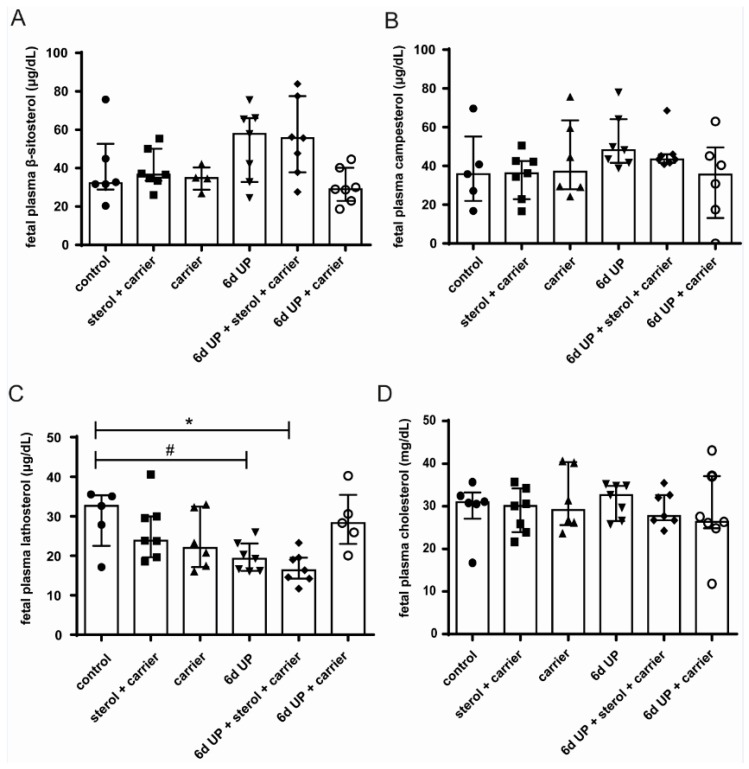
Circulatory β-sitosterol, campesterol, lathosterol (cholesterol precursor), and cholesterol levels in lambs of 133 d GA (preterm delivery). No significant differences were found for β-sitosterol (**A**) and campesterol (**B**) in all treatment groups. (**C**) UP exposure with and without plant sterol treatment reduced lathosterol plasma concentrations compared to controls. (**D**) Plasma cholesterol concentrations were not affected by UP exposure and/or plant sterol treatment. * *p* < 0.05; # 0.05 < *p* < 0.10. Abbreviations: GA, gestational age; UP, *U. parvum*.

**Figure 7 nutrients-11-00968-f007:**
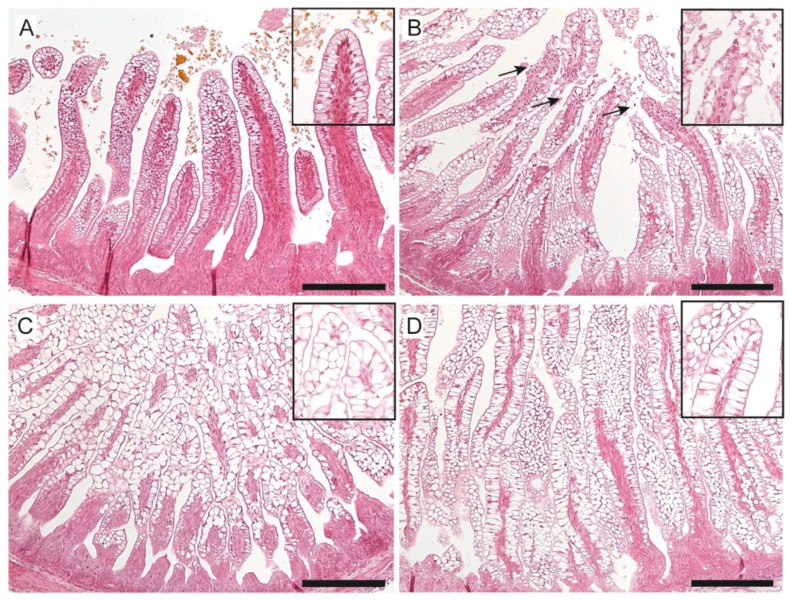
Evaluation of morphological changes and enterocyte vacuolization in the distal ileum of preterm lambs by H&E staining in control (**A**), UP-exposed (**B**), UP + carrier-exposed (**C**), and UP + sterol + carrier-exposed fetal lambs (**D**). Injury of the villi tips (arrows) was observed after 6 d UP exposure, which was prevented by carrier and/or sterol + carrier treatment. The size of the vacuoles within the enterocytes was increased in all experimental groups and most profoundly in UP + carrier and UP + sterol + carrier animals. Scale bar indicates 200 µm. Abbreviations: H&E, hematoxylin and eosin; UP, *U. parvum*.

**Figure 8 nutrients-11-00968-f008:**
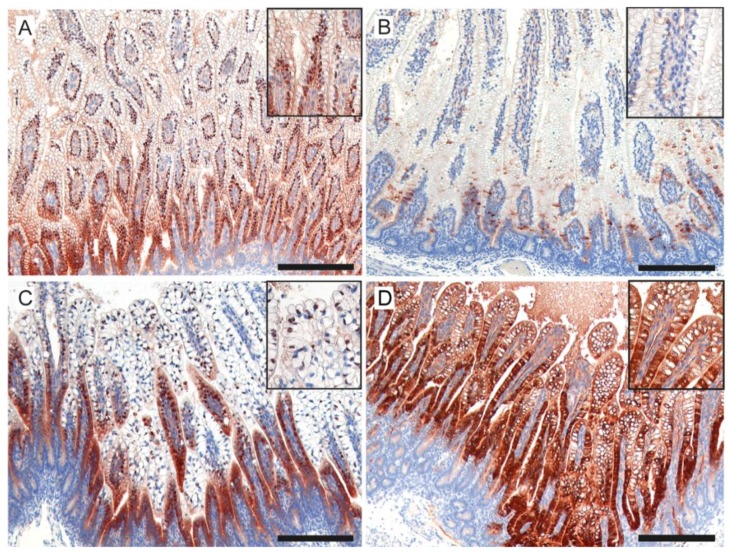
Expression pattern of I-FABP in the ileal gut. (**A**) In control fetuses, I-FABP expression was present along the crypt–villi axis. (**B**) This pattern was disrupted in UP-exposed lambs, where I-FABP expression was more localized in the intestinal crypts. (**C**) The I-FABP expression pattern was restored in most preterm lambs treated with the carrier, without and with (**D**) plant sterols with a pattern comparable to that seen in the control intestinal samples. Scale bar indicates 200 µm. Abbreviations: I-FABP, intestinal fatty acid binding protein; UP, *U. parvum*.

**Figure 9 nutrients-11-00968-f009:**
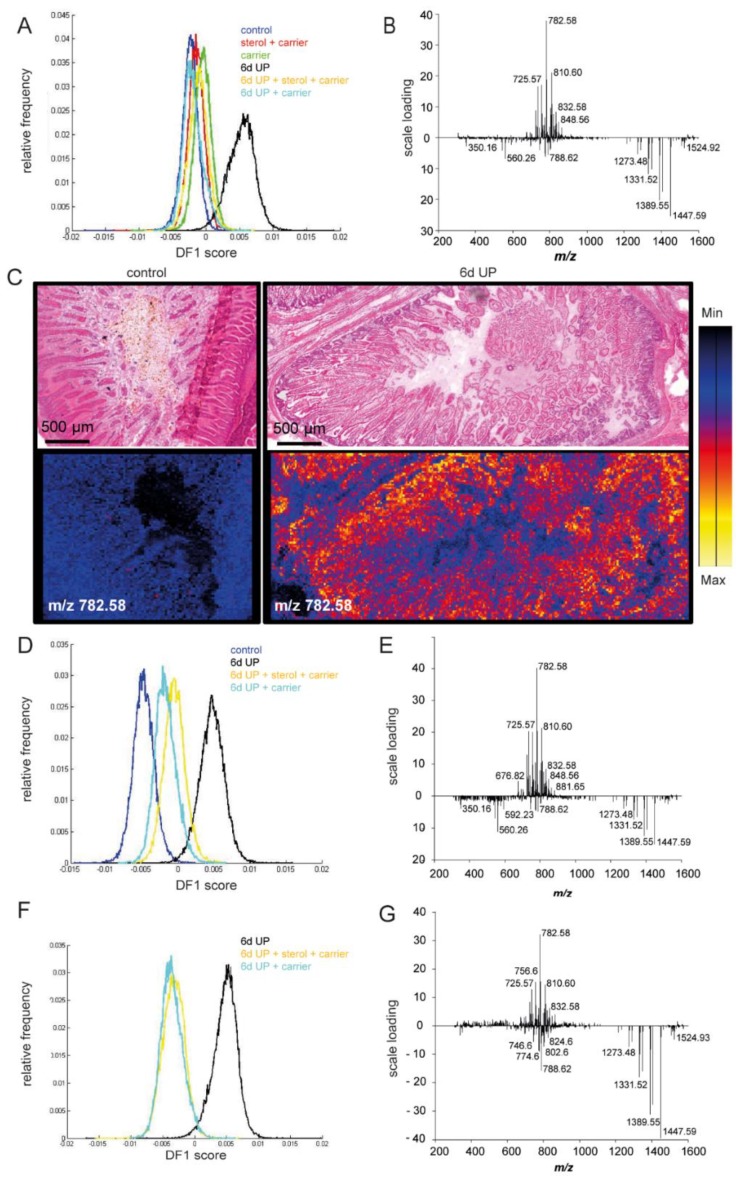
Matrix-assisted laser desorption ionization mass spectrometry imaging performed in positive ion mode followed by principal component analysis. The first discriminant function is shown. (**A**) Intestinal lipid composition was changed after 6 d UP exposure compared to the other treatment groups, (**B**,**C**) with increased presence of different PCs and SMs in the 6-d UP group, including *m*/*z* 782.58 PC 34:1 [M + Na]^+^, *m*/*z* 810.60 PC 36:1 [M + Na]^+^, and *m*/*z* 725.57 SM 34:1 [M + Na]^+^. (**D**–**G)** These UP-induced changes in the intestinal lipidome were partially normalized by plant sterols and/or the carrier. The carrier (2-hydroxypropyl-β-cyclodextrin), characterized by *m*/*z* 1331.52 [M + Na]^+^, *m*/*z* 1389.55, and *m*/*z* 1447.59, was detected in the intestinal epithelium of the 6-d UP + sterol + carrier and the 6-d UP + carrier groups. Abbreviations: PCs, phosphocholines; SMs, sphingomyelins; UP, *U. parvum*.

**Figure 10 nutrients-11-00968-f010:**
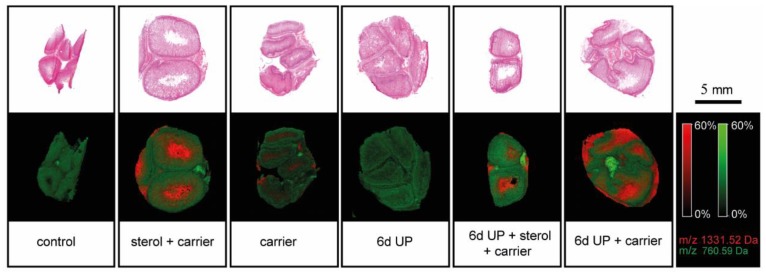
Presence of the carrier, indicated by *m*/*z* 1331.52 [M + Na]^+^, was detected in the intestinal lumen in the carrier, 6-d UP + carrier, sterol + carrier, and 6-d UP + sterol + carrier groups, but not in the control and 6-d UP groups: *m*/*z* 760.59 [M + H^+^ was a homogenously distributed PC lipid present in the intestinal epithelial layer. Scale bar indicates 5 mm. Abbreviations: PCs, phosphocholines; UP, *U. parvum*.

**Figure 11 nutrients-11-00968-f011:**
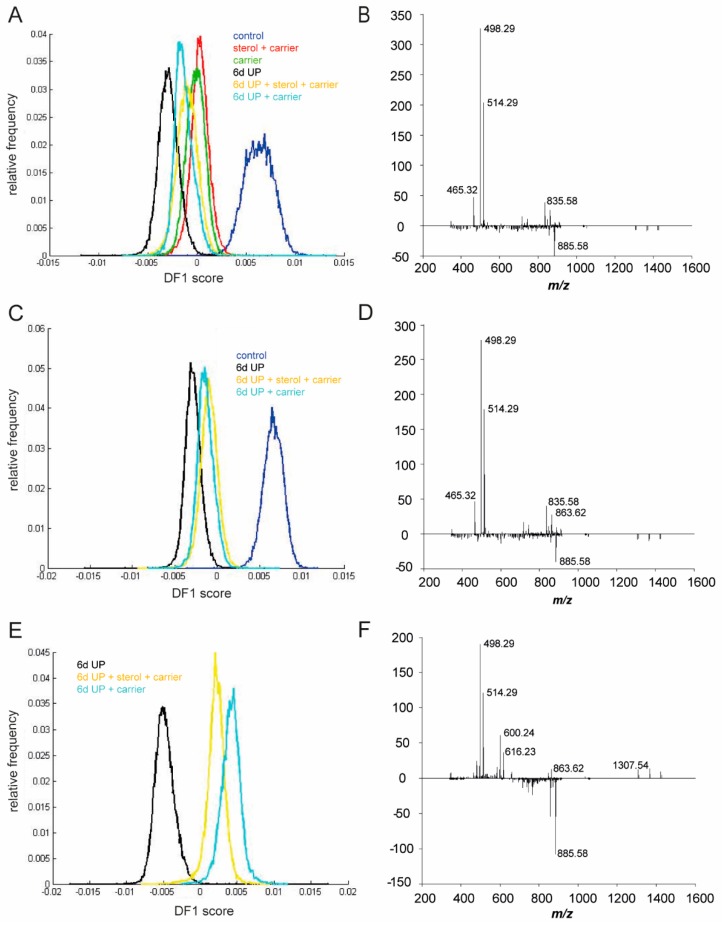
Matrix-assisted laser desorption ionization mass spectrometry imaging performed in negative ion mode followed by principal component analysis. The first discriminant function is shown. (**A**) 6-d UP group animals showed an altered lipid profile compared to the other treatment groups, (**B**) with a higher abundance of *m*/*z* 885.58 PI 38:4 [M−H]^−^ and a reduced presence of *m*/*z* 498.29 [M−H]^−^ dihydroxy BA TCDA/TCDCA and *m*/*z* 514.29 [M−H]^−^ trihydroxy BA TCA. (**C**–**F**) In the 6-d UP + sterol and 6-d UP + sterol + carrier groups, UP-induced lipidome and BA changes were partially prevented with a profile comparable to the controls. Abbreviations: BA, bile acid; PI, phosphatidylinositol; TCA, taurocholic acid; TCDA, taurodeoxycholic acid; TCDCA, taurochenodeoxycholic; UP, *U. parvum*.

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
