# Peer review of "Protection of the Ovine Fetal Gut against Ureaplasma-Induced Chorioamnionitis: A Potential Role for Plant Sterols"

_nutrients, 2019, doi:10.3390/nu11050968_

Round 1
Reviewer 1 Report
The study aimed to investigate the anti-inflammatory effects of intra-amniotically delivered plant-based sterols in fetal lambs exposed to ureaplasma. Overall this is a well-designed study with several control groups albeit with some limitations. The authors state that a group size of 50 was based on power analysis, however they did not mention the primary outcome for the power analysis nor the data used for the calculations. A further limitation was the presentation of data. Figures are presented as scatterplots overlaid with a column and error bars based on mean and standard error. This provides a detailed presentation of data, although it would be more appropriate that the overlay be a median or box and whiskers plot as the small number of data points are treated as non-parametric. The authors imply that plant sterols can reach the AF if consumed orally. It is disappointing that an additional group was not included to determine if oral consumption of plant sterols can similarly increase AF sterol levels with similar anti-inflammatory effects. The claim that this is a translational study is not supported. IA administration of plant sterols is not a reasonable practice, there is only an inference that orally consumed plant sterols will reach the AF and there is no evidence that orally consumed plants sterols can reach therapeutic concentrations in the AF. The authors should adjust their discussion and conclusions to frame the findings of this study in a more reasonable context.
Author Response
Dear reviewer,
Thank you very much for offering us the opportunity to submit a revised version of our manuscript ID Nutrients-477347 entitled “Protection of the ovine fetal gut against Ureaplasma-induced chorioamnionitis; a potential role for plant sterols”.
We thank you for your interest in our work and appreciation of our study to address the impact. We are grateful for the valuable comments that we feel have improved our manuscript.
Please find enclosed a revised version of the manuscript and a point to point response to the comments of the reviewers.
Q: The authors state that a group size of 50 was based on power analysis, however they did not mention the primary outcome for the power analysis nor the data used for the calculations.
R: The primary outcome for the power analysis was intestinal inflammation. We have adjusted this in the manuscript lines 97-98. Group sizes were based on previous studies, with a power of 80% and alpha 0.05 significant different inflammation in the gut between the control and experimental group of 20% (δ=20) with a SD of 12% (σ=12) can be detected with a sample size of n = 6 ( n = 15.7 * (12/20)2 = 5.7). Taken into account a loss of 2 animals per experimental group, the total group size was n=9 per group for the sterol and UP + sterol group and n=8 for the other experimental groups. In case the reviewer prefers to incorporate the full details on the power analysis in the manuscript, this can be integrated at line 98.
Q: A further limitation was the presentation of data. Figures are presented as scatterplots overlaid with a column and error bars based on mean and standard error. This provides a detailed presentation of data, although it would be more appropriate that the overlay be a median or box and whiskers plot as the small number of data points are treated as non-parametric.
R: The figures have been adjusted, now showing median and interquartile range instead of mean and standard error of the mean (line 232-233).
Q: The authors imply that plant sterols can reach the AF if consumed orally. It is disappointing that an additional group was not included to determine if oral consumption of plant sterols can similarly increase AF sterol levels with similar anti-inflammatory effects. The claim that this is a translational study is not supported. IA administration of plant sterols is not a reasonable practice, there is only an inference that orally consumed plant sterols will reach the AF and there is no evidence that orally consumed plants sterols can reach therapeutic concentrations in the AF. The authors should adjust their discussion and conclusions to frame the findings of this study in a more reasonable context.
R: We thank the reviewer for these comments. Based on the remark of the reviewer, the rationale behind our choice to administer plant sterols IA has been further explained in the materials and methods section (lines 103-112). Moreover, changed the translational study in a proof of concept study (line 466) and admit that the pharmacokinetics for the fetus following maternal intake of plant sterols during pregnancy is unknown, by incorporating a limitation sentence regarding this pharmacokinetics in the conclusion section (line 469-470).
Reviewer 2 Report
Dear Editor, the present article is well designed and well written.
Author Response
Dear reviewer,
Thank you very much for offering us the opportunity to submit a revised version of our manuscript ID Nutrients-477347 entitled “Protection of the ovine fetal gut against Ureaplasma-induced chorioamnionitis; a potential role for plant sterols”.
We thank you for your interest in our work and appreciation of our study to address the impact. We are grateful for the valuable comments that we feel have improved our manuscript.
Please find enclosed a revised version of the manuscript.